# Trade-Offs between Temperature and Fitness in *Euschistus heros* (Fabricius) (Hemiptera: Pentatomidae): Implications for Mass Rearing and Field Management

**DOI:** 10.3390/insects14050448

**Published:** 2023-05-10

**Authors:** Ana Paula Frugeri Barrufaldi, Rafael Hayashida, William Wyatt Hoback, Leon G. Higley, Jose Romario de Carvalho, Regiane Cristina de Oliveira

**Affiliations:** 1Crop Protection Department, Campus of Botucatu, São Paulo State University, Botucatu 18610-034, São Paulo, Brazil; 2Department of Entomology and Plant Pathology, Oklahoma State University, Stillwater, OK 74078, USA; 3School of Natural Resources, University of Nebraska, Lincoln, NE 68583-0760, USA; 4State Department of Education, State of Espírito Santo, Jerônimo Monteiro 29550-000, Espirito Santo, Brazil

**Keywords:** brown stink bug, *glycine max*, life table multiple decrements, economic threshold

## Abstract

**Simple Summary:**

The brown stink bug, *Euschistus heros*, is a key pest affecting soybean crops in Brazil and other South American countries. Temperature plays a crucial role in insect development and reproduction; however, little is known about temperature on *E. heros*’ fitness. This study examined the effects of both constant and fluctuating temperatures on *E. heros*’ biological characteristics across three generations. Development did not occur at constant temperatures of 19 °C and 34 °C, and at 31 °C, there was no reproduction. Development duration was faster at warmer constant temperatures, but fluctuating temperatures decreased *E. heros*’ egg viability. This research allowed the development of temperature base and degree day requirements for each *E. heros* stage and revealed that about 40% mortality occurs during molting, with the highest mortality between first and second-stage nymphs. Together, these data improve the management of *E. heros* and benefit the mass production of egg parasitoids used for its control.

**Abstract:**

The brown stink bug, *Euschistus heros* (Fabricius, 1798) (Hemiptera: Pentatomidae), is one of the most abundant soybean stink bug pests in Brazil. Temperature is a key factor that affects its development and reproduction, and fluctuating temperatures may impact the development and reproduction of *E. heros* differently from those under constant temperatures. The objective of this study was to evaluate the influence of constant and fluctuating temperature on the biological characteristics of *E. heros* in three successive generations. Treatments consisted of six constant temperatures (19 °C, 22 °C, 25 °C, 28 °C, 31 °C and 34 °C) and four fluctuating temperatures (25:21 °C, 28:24 °C, 31:27 °C, and 34:30 °C) evaluated for three successive generations. Second-stage nymphs were evaluated daily, and after they reached the adult stage, they were separated by sex, and the individual weight (mg) and pronotum size (mm) were recorded. After pair formation, eggs were collected to evaluate the pre-oviposition period, total number of eggs, and egg viability. The duration of the nymphal stage was reduced with an increase in both constant and fluctuating temperatures; however, at constant temperatures of 19 °C, 31 °C and 34 °C and fluctuating temperatures of 28:24 °C, there was no reproduction in adults. The base temperature and total degree day requirement for nymphal development were 15.5 °C and 197.4 dd, respectively. Pre-oviposition period (d), number of eggs per female, and viability of eggs (%) were affected by temperature across the generations. The multiple decrement life table analysis revealed that mortality was highest during the molting of the second-stage nymphs. These findings have important implications for *E. heros*’ laboratory mass-rearing programs and for its management in fields.

## 1. Introduction

The brown stink bug, *Euschistus heros*, native to neotropical regions, is an important pest of soybean crops in South America, causing significant yield losses. Although it was considered uncommon until the 1970s [1], it is now one of the most abundant stink bug pests affecting major commodities in Brazil [2,3,4]. It has been found feeding on 21 different plant species, and the widespread adoption of no-tillage cultivation systems and the practice of multiple cropping may have favored its spread [5]. The expansion of soybean toward warmer regions, such as the Brazilian Cerrado and Amazon [6], may have also impacted *E. heros*’ distribution and its dynamics.

The brown stink bug utilizes its stylets to feed on various structures of the soybean plant, but its feeding on the pod can have significant consequences for soybean production, including reduced yield, altered grain quality parameters, and changes in oil and protein content of the seeds during pod development and seed filling [7,8]. In the field, *E. heros* undergoes five nymphal instars, but its injury to soybeans is only significant after the third instar [4,9]. Spraying chemical insecticides is the primary management tactic adopted by soybean growers [10], but it has several consequences, including the selection of resistant pest populations, reduction of biological control agents, and outbreaks of secondary pests [11,12]. To minimize these effects, Integrated Pest Management (IPM) recommends the adoption of diverse approaches, including the use of economic thresholds (ETs) and biological control methods, when available [3].

The ET is defined as the pest density at which the management tactic should be applied to prevent its population from reaching the economic injury level, that is, the lowest level of pest population density or damage that will cause economic losses equal to the cost of implementing pest control measures [13]. The ET is determined by a combination of factors, including the value of the crop, the cost and efficiency of control measures, and the pest’s biology and behavior.

Despite the well-established ET for brown stink bugs on soybean, which recommends treatment when there are two stink bugs per meter (third to fifth instar nymphs and/or adults) [4,7], soybean growers still express concerns about stink bug population growth in the field, particularly when numerous eggs and small nymphs up to the second instar are detected during sampling. As the species has spread and because climate change is increasing temperatures, it is crucial to investigate the impact of temperature on the mortality of each stink bug instar, as this information is critical for both informing growers’ decision-making and generating data for modeling future pest outbreaks.

The investigation of the impact of temperature on the biology of *E. heros* is necessary not only for determining the ET of soybean fields, but also for laboratory mass rearing of the brown stink bug. *Euschistus heros*’ mass rearing has various applications, among them, for the biological control programs using egg parasitoid wasps [14,15,16]. The utilization of egg parasitoids such as *Telenomus podisi* Ashmead 1893 (Hymenoptera: Platygastridae) as a management tool for *E. heros* has demonstrated promising outcomes [17,18]. However, the wide adoption of egg parasitoids as a biological control tool for *E. heros* requires a large-scale production of their hosts with both quality and quantity [19]. Considering the vast area of soybean cultivation, the investigation of the impact of temperature on *E. heros* biology can also aid in overcoming production limits and promote the widespread adoption of utilizing egg parasitoids.

Temperature plays a critical role in insect development and reproduction, and its effects on biological processes in insects are non-linear [20]. This means that fluctuating temperatures can lead to different physiological, life history, and ecological outcomes compared to those observed under constant temperatures [20]. Prior research has examined the influence of fluctuating temperatures on a broad range of diverse insects, including biological control agents [21,22], aquatic insects [23] and the brown marmorated stink bug, *Halyomorpha halys* (Stål, 1855) (Hemiptera: Pentatomidae) [24]. However, for the brown stink bug, no scientific literature has investigated its development and reproduction under different constant or fluctuating temperatures. Therefore, the objective of this study was to assess the impact of constant and fluctuating temperatures on the biological characteristics of *E. heros*; utilize the gathered data to establish temperature requirements for each instar; develop a decrement life table, with the aim of providing fundamental knowledge to enhance soybean IPM programs, including laboratory mass-rearing programs and field management strategies.

## 2. Materials and Methods

The laboratory rearing of *E. heros* originated from insects collected in a soybean field at the Experimental Farm Lageado of the Faculty of Agronomic Sciences of UNESP-Botucatu, São Paulo State, Brazil (geographic coordinates 22°48′19.5″ S 48°25′38.4″ W). The founding unit of the rearing was comprised of 156 insects, including 70 males and 86 females, and these insects were maintained for three generations until the beginning of the experiment.

Nymph and adult *Euschistus heros*’ were reared in the laboratory following the methodology described by Silva et al. [25]. The insects were kept in climate-controlled rooms with a temperature of 25 ± 2 °C, relative humidity (RH) of 70 ± 10%, and photophase of 14 h. The egg clutches were placed in 11 cm × 11 cm × 3.5 cm Gerbox^®^ plastic boxes and kept under controlled conditions at a temperature of 25 ± 1 °C, photophase of 14 h, and RH of 60 ± 10% until hatching. Four plastic microtubes containing cotton soaked in water were placed inside each plastic box to maintain humidity inside the container.

Stink bugs from the second instar to the adult stage were placed individually in 60 mm × 10 mm Petri dishes, each with a 5 mm × 5 mm hole in the lid of the plate, covered with fine metal mesh to allow air to enter. The bottom of each Petri dish was lined with a piece of filter paper, which was replaced as needed. The stink bugs were provided a diet composed of ligustrum (*Ligustrum lucidum*), common bean (*Phaseolus vulgaris*), soybeans (*Glycine max*), and peanuts (*Arachis hypogea*), which was placed in each plate along with an Eppendorf^®^ plastic microtube with cotton soaked in water [26] and replaced three times per week.

There were ten treatments used to evaluate the impact of temperature on the biological characteristics of *E. heros*: six constant temperatures (19 °C, 22 °C, 25 °C, 28 °C, 31 °C, and 34 °C, each ± 1 °C), and four fluctuating temperatures (day:night: 25:21 °C, 28:24 °C, 31:27 °C, 34:30 °C, all ± 1 °C) with a 14 h photophase RH 60 ± 10% in BOD chambers for all treatments, for three successive generations. Temperature and humidity inside each chamber were measured weekly with an Incoterm thermo-hygrometer, and thermal adjustments were made if necessary.

In the first generation (F0), there were six pseudo-replicates of 20 nymphs, totaling 120 individuals, in each treatment. They were placed in Petri dishes in climate-controlled chambers for each temperature tested. Because *E. heros* could not survive and reproduce at all tested temperatures, the number of pseudo-replications (n) was based on results from the previous generation. In the second generation (F1), there were 8 treatments with 16 pseudo-replicates each, and in the third generation (F2), there were 8 treatments with 5 pseudo-replicates each. The experimental unit was established as climate-controlled chambers, with each Petri dish being considered a pseudo-replicate. The experimental design comparing temperatures was pseudo-replicated, as the experimental unit was the chamber, not the individual petri dishes. As a result, a conservative statistical approach was taken when making treatment comparisons.

The parameters that were evaluated during the nymphal phase were the duration and survival of each instar, as well as the total nymphal cycles for all three generations. Twenty-four hours after the emergence of adults, the insects were weighed (mg) using a Shimadzu precision analytical scale (model Ay220) and photographed under a stereoscopic microscope at 40× magnification using the Leica Application Suite software to measure the pronotum size (mm).

The adults were then separated by sex to determine the sex ratio (females: males) and, subsequently, allowed to form couples. The first generation (F0) had 9 treatments with 6 replicates in the adult phase, and the second and third generations each had 8 treatments with 6 replicates.

Individual mating pairs were transferred to 11 cm × 11 cm × 3.5 cm Gerbox^®^ plastic boxes along with cotton for oviposition, which occurs preferentially in this substrate [26]. Offspring were kept at the treatment temperature for three generations to determine the pre-oviposition period, total number of eggs, and egg viability. In the first generation, 9 treatments with 5 replicates; in the second generation, 8 treatments with 3 replicates, and in the third generation, 8 treatments with 4 replicates could be evaluated for all these biological parameters.

The containers with the adult pairs were observed daily until the first eggs were laid; then, the eggs were collected three times per week, with a minimum interval of two days between collections. Each pair remained in the container for a period of 50 days after the first egg laying, and then surviving individuals were discarded.

### Statistical Analysis

The experiment was carried out in a completely randomized design for nymph stages, and for adult stages, it was carried out in a 2-factor factorial design, in an arrangement of sex × temperature. The parameters evaluated for the nymphal stage were duration and survival of each instar and total. For the adult stage, sex ratio, weight, and pronotum size were evaluated. For couples pre-oviposition period, the total number of eggs and egg viability were evaluated. Because of the pseudo-replication of temperature, Tukey’s post-hoc test was employed, which is a well-established and relatively conservative method for separating treatment means [27].

The variables survival, sex ratio and viability of the eggs were analyzed using a generalized linear model (GLM) with residual Bernoulli distribution because data vary between 0 and 1 for sex ratio and from 0 to 100 for survival and viability of the eggs. The variable total number of eggs was analyzed using a GLM with negative binomial residual distribution because these are count data. Analyzes were performed using the ExpDes.pt package in the R computing environment [28].

In addition, the base temperature and degree day (dd) requirement for each instar were calculated. The base temperature and dd requirement are measures used to estimate the growth and development rates of insects based on temperature exposure.

The study also examined the impact of molting mortality on stage-specific and cumulative mortality using a multiple decrement life table. A multiple decrement life table is a type of life table that considers different causes of mortality, such as age-specific mortality and cause-specific mortality, to estimate the survival and mortality rates of a population [29]. In this case, the study examined the effect of molting mortality on each instar of the stink bug population.

## 3. Results

The duration (days) of the nymphal stage and each instar of *E. heros* were reduced with an increase in both constant and fluctuating temperatures (Table 1). At a constant temperature of 19 °C, the nymphal stages of the F0 generation were prolonged, and there were no survivors in the fifth instar, resulting in no further generations (Table 1). Higher fluctuating temperatures also resulted in a shorter nymphal stage, with the shortest cycle around 14 days at 34:30 °C. There was a small variation in the nymphal duration between generations under both constant and fluctuating temperatures. At 28 °C, the difference between F0 and F1 was less than 2 days, but there was no significant difference between F0 and F2 (Table 1).

The base temperature for the nymphal development (second to fifth instar) was 15.5 °C, with slight variation within each instar. The lowest base temperature was observed during the fifth instar development (13.5 °C), while the highest was recorded during the fourth instar development (15.2 °C; Table 2). The total dd requirement for nymphal development was 197.4 dd. The fifth instar required the most dd (84.9), and the third instar required the fewest dd (43.7; Table 2).

During the nymphal stage, the survival rate (%) was substantially higher at F2 for both constant and fluctuating temperatures, except for the fluctuating temperature treatment of 25–21 °C, where F2 had 25% fewer survivors (Table 3). The lowest and the highest constant temperatures tested (19 °C and 34 °C) prevented the nymphal development from F1. The survival rate (%) of the second instar nymphs of the F2 and the fifth instar nymphs of the F1 were greatly reduced at 31 °C (Table 3). According to the multiple decrement life table, more than 40% of the mortality was caused by molting (Table 4). About half of these deaths occurred when insects molted from the second to third instar.

The weight and pronotum size of adults were found to be significantly different between sexes in all three generations. Females had higher weight (mg) and pronotum size (mm) compared to males (Figure 1 and Figure 2). A constant temperature of 22 °C produced both females and males with larger pronotum sizes, while fluctuating temperatures of 25:21 °C and 28:24 °C produced insects with smaller pronotum sizes in all three generations (Table 5). The sex ratio, however, had no significant difference among all the constant and fluctuating temperatures tested or among generations (Table 6).

Pre-oviposition period (d), number of eggs per female, and viability of eggs (%) were affected by temperature and across the generations (Table 7). There was an inverse correlation between temperature and pre-oviposition period (d); as temperature increased, the pre-oviposition period decreased. Furthermore, *E. heros* adults did not reproduce at the temperatures of 19 °C, 31 °C, and 34 °C, or at the fluctuating temperature of 25:21 °C.

The pre-oviposition period was also affected by the generation, with F2 generation females having a longer pre-oviposition period (d) compared to F1 and F1 females having a longer pre-oviposition period than F0 (Table 7). Although this occurred at all temperatures tested, at 28 °C, this difference was even more evident, with F2 females spending about 13 times longer in the pre-oviposition period compared to F0 females (2.31 ± 0.41 d for F0 and 29.48 ± 0.45 d for F2).

In contrast, the number of eggs per female was lower in F0 females, especially at 22 °C, 28 °C, and with the fluctuating temperature of 31:27 °C (Table 6). Females at a constant temperature of 25 °C had a higher mean number of eggs (ranging from 347.00 ± 57.97 to 560.52 ± 28.36 eggs per female). The females in the fluctuating temperature of 34:30 °C had the lowest mean number of eggs overall (174.81 ± 40.96 for F0, 197.20 ± 25.65 for F1, and 257.30 ± 49.72 for F2). All the fluctuating temperatures tested had lower egg viability (%) when compared with constant temperatures across all generations tested, with the lowest value found at the fluctuating temperature of 34:30 °C, where less than 9% of the eggs were viable (Table 7).

## 4. Discussion

In the present work, we identified the lower and upper temperature thresholds for the development of *E. heros*, with survival being compromised below a constant temperature of 19 °C and above 34 °C. Although previous research has evaluated the effects of temperature on stink bugs [24,30,31,32,33], this study is the first to assess the effects of constant and fluctuating temperatures on three generations under laboratory conditions. Our findings reveal an inverse relationship between temperature and nymphal development duration and a direct relationship between temperature and survival (%), especially under constant temperatures. Similar results have been reported in the scientific literature for the brown marmorated stink bug, *H. halys* [24]. Such a trade-off is crucial to consider for improving the mass rearing of *E. heros* in laboratory settings where there is a demand for large quantities of insects within a short period of time. Acceleration of *E. heros* development can occur with increasing temperature, but it may compromise survival, highlighting the importance of temperature control in mass-rearing programs.

In addition to survival, it is crucial to consider reproductive parameters such as the pre-oviposition period, egg production, and viability when mass-rearing *E. heros* in laboratory settings. Our study revealed that the pre-oviposition period of *E. heros* was prolonged in F1 and F2 generations at all viable temperatures, indicating a potential laboratory-induced effect on the population reared under controlled conditions. Additionally, the use of fluctuating temperatures for mass rearing may not be suitable for *E. heros*, as we observed a significant decrease in egg viability under such conditions. These findings highlight the importance of carefully selecting appropriate laboratory techniques for successful mass rearing of *E. heros*, particularly for producing insects of high quantity for various applications, including bioecology studies [5], insecticide resistance tests [34], and biological control programs [19,35,36].

Along with the reproductive and developmental parameters, the study of temperature impact on insect fitness should also measure morphometric parameters. The results of this study suggest that while temperature did not have a significant impact on the weight of the brown stink bug, pronotum size was reduced in both sexes across three generations with an increase in constant temperature. Pronotum size is an important measurement in evaluating environmental stress and other physiological impacts to stink bugs, as it may indicate responses related to diapause and seasonal form modifications [37].

The results of this study also have important implications for the management of *E. heros* in soybean fields. In field conditions, where temperatures fluctuate throughout the day, the fluctuating temperature is likely to have a greater impact on the fitness of *E. heros* compared to constant temperatures. The reduced egg viability observed under fluctuating temperatures suggests that *E. heros* populations in soybean fields may experience reduced reproductive success and potentially lower population growth rates.

Furthermore, this study provided the temperature base and the dd requirement for each instar. This information can be used to improve pest management strategies for the brown stink bug populations [30]. For example, we can use degree day calculations to predict when the population of *E. heros* will reach the third instar (the instar where stink bugs start damaging soybean plants [4,38]) and help growers to plan interventions, such as insecticide applications or other management strategies, to limit populations of *E. heros* before they reach the ET.

In addition to mortality associated with temperatures, the multiple decrement life table analysis used in this study revealed that the mortality of *E. heros* was highest during the molting of the second instar. This information is crucial for stink bug management, as it indicates that more than 20% of second instar insects died under controlled conditions from the natural process of molting and will, therefore, not reach the stage that affects soybean yield. It is important to point out that, in addition to the death caused by molting, stinkbug populations in fields are limited by natural enemies, such as predators and parasitoids, diseases and abiotic factors, such as rain. Some *E. heros* parasitoids can benefit from warmer temperatures, increasing their efficiency in the field [39].

Further tests to confirm these results are warranted. All of our experiments were conducted under well-controlled laboratory conditions, which may not suitably represent the complex and dynamic environmental factors that *E. heros* experiences in the field. In the field, it is likely that adults seek microclimate refuges and thus, laboratory results under constant conditions may not represent the behavior and development of the stink bug under natural conditions. Further studies are necessary to examine other abiotic factors, such as humidity, precipitation, and varying light regimes, which might also influence the development and survival of the brown stink bug. Furthermore, future research should also address the effect of fluctuating temperatures on its natural enemy, the egg parasitoid *T. podisi*, to provide a better optimized potential for biological control programs.

## 5. Conclusions

The present study found that there is no development at constant temperatures of 19 °C and 34 °C and no reproduction at 31 °C. An inverse relationship was observed between temperature and nymphal development duration at remaining temperatures. The study also found that temperature had a significant impact on the survival and pre-oviposition period of *E. heros*. Furthermore, it provided degree day requirements for each instar and identified molting as a critical factor in population mortality (more than 40%), shedding light on an important factor of *E. heros*’ natural mortality, the ecdysis process. Despite the limitations of this study, these findings have important implications for *E. heros* laboratory mass-rearing programs, guiding the adoption of better temperature regimes to produce eggs for mass-rearing egg parasitoids and for its management in the field. Moreover, this sheds light on an important factor of *E. heros*’ natural mortality, the ecdysis process.

## Figures and Tables

**Figure 1 insects-14-00448-f001:**
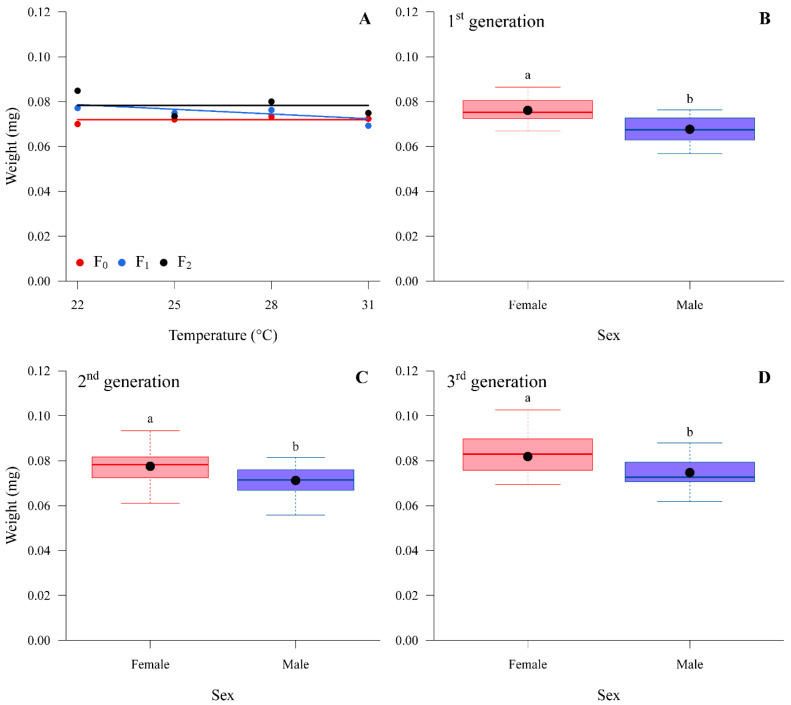
Weight (mg) of adult *Euschistus heros* submitted to different temperatures constant for three generations, under laboratory conditions; (**A**) boxplot of adult females and males of *E. heros* of the 1st generation; (**B**) adult females and males of *E. heros* of the 2nd generation; (**C**) and adult females and males of *E. heros* of the 3rd generation (**D**). Boxplot represents its five-number summary: minimum, first quartile (Q1), median (Q2), third quartile (Q3), and maximum, and the black dot represents the average value. Boxplot accompanied by different letters differ from each other by the F test.

**Figure 2 insects-14-00448-f002:**
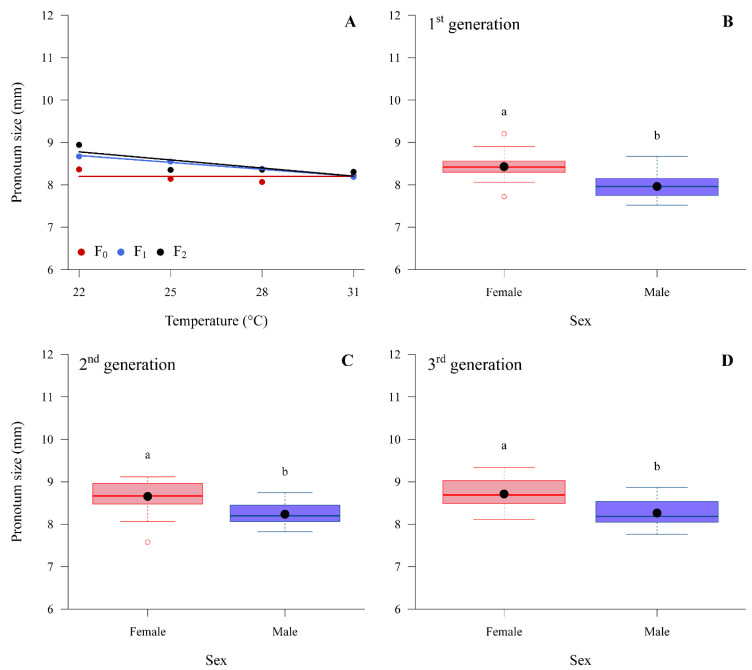
Pronotum size (mm) of adult *Euschistus heros* submitted to different temperatures constant for three generations, under laboratory conditions; (**A**) boxplot of adult females and males of *E. heros* of the 1st generation (F = 36.73, *p*-value < 0.0001); (**B**) adult females and males of *E. heros* of the 2nd generation (F = 30.12, *p*-value < 0.0001); (**C**) and adult females and males of *E. heros* of the 3rd generation (F = 66.747, *p*-value < 0.0001). (**D**) Boxplot represents its five-number summary: minimum, first quartile (Q1), median (Q2), third quartile (Q3), and maximum, and the black dot represents the average value. Boxplot accompanied by different letters differ from each other by the F test.

**Table 1 insects-14-00448-t001:** Duration (days ± SE) of the *Euschistus heros*’ instars submitted to different temperatures constant and alternating for three generations under laboratory conditions. Means followed by the same letter, lowercase in column (between generations) and uppercase in the line, do not differ from each other by Tukey test (α = 0.05).

Generation ^1^	Temperatures									
	Constant						Fluctuating			
	19 °C	22 °C	25 °C	28 °C	31 °C	34 °C	(25–21 °C)	(28–24 °C)	(31–27 °C)	(34–30 °C)
	2nd instar									
F_0_	11.07 ± 0.35 A	5.88 ± 0.10 bB	4.82 ± 0.12 bCD	2.97 ± 0.10 bFG	2.60 ± 0.08 bG	2.47 ± 0.01 G	5.07 ± 0.08 aC	4.10 ± 0.21 aDE	3.71 ± 0.14 aEF	2.38 ± 0.06 aG
F_1_	-	6.97 ± 0.17 aA	4.86 ± 0.09 bB	3.72 ± 0.12 aCD	3.21 ± 0.05 abCD	-	4.84 ± 0.45 aB	4.68 ± 0.68 aB	3.68 ± 0.11 aBC	3.02 ± 0.31 aD
F_2_	-	6.18 ± 0.06 bA	5.50 ± 0.22 aA	3.87 ± 0.15 aB	3.73 ± 0.29 aB	-	5.73 ± 0.04 aA	3.15 ± 0.33 aB	3.71 ± 0.48 aA	3.13 ± 0.43 aB
	3rd instar									
F_0_	12.50 ± 0.48 A	6.19 ± 0.24 aB	5.43 ± 0.22 aB	3.57 ± 0.06 aDE	3.08 ± 0.08 aEF	2.61 ± 0.08 F	5.37 ± 0.24 aBC	4.43 ± 0.31 aCD	3.71 ± 0.08 aDE	3.17 ± 0.16 aEF
F_1_	-	6.28 ± 0.20 aA	4.12 ± 0.08 bBC	3.17 ± 0.14 abDE	2.80 ± 0.16 abE	-	4.76 ± 0.13 aB	4.26 ± 0.30 aBC	3.68 ± 0.14 aCD	3.03 ± 0.20 aDE
F_2_	-	5.34 ± 0.11 bA	5.23 ± 0.22 aA	3.07 ± 0.15 bBC	2.48 ± 0.13 bC	-	4.74 ± 0.08 aA	3.46 ± 0.19 aB	3.71 ± 0.13 aB	3.22 ± 0.16 aB
	4st instar									
F_0_	17.21 ± 0.62 A	7.28 ± 0.16 aB	6.02 ± 0.24 aCD	3.77 ± 0.07 aFG	3.17 ± 0.15 aFG	2.97 ± 0.07 G	6.83 ± 0.19 aBC	5.42 ± 0.28 aDE	4.29 ± 0.13 aEF	3.86 ± 0.16 aFG
F_1_	-	6.72 ± 0.26 aA	4.87 ± 0.32 bB	3.78 ± 0.20 aC	2.87 ± 0.08 aD	-	5.90 ± 0.40 aA	5.06 ± 0.33 aB	3.98 ± 0.31 aC	3.27 ± 0.11 aCD
F_2_	-	5.89 ± 0.06 bA	5.79 ± 0.13 abA	3.45 ± 0.21 aCD	3.08 ± 0.15 aD	-	3.98 ± 0.15 bBCD	4.63 ± 0.36 aB	4.05 ± 0.15 aBC	3.84 ± 0.20 aBCD
	5st instar									
F_0_	-	10.15 ± 0.34 abA	8.11 ± 0.08 aBC	6.18 ± 0.20 aD	5.09 ± 0.12 aEF	4.52 ± 0.14 F	8.68 ± 0.16 aB	7.76 ± 0.07 aC	7.34 ± 0.16 aC	5.88 ± 0.06 aDE
F_1_	-	10.37 ± 0.14 aA	8.03 ± 0.12 aB	4.95 ± 0.13 bD	5.04 ± 0.21 aD	-	10.21 ± 0.37 aA	6.33 ± 0.29 bC	6.80 ± 0.33 aC	5.10 ± 0.12 bD
F_2_	-	9.30 ± 0.31 bA	7.19 ± 0.16 bB	5.75 ± 0.11 aCD	4.71 ± 0.20 aD	-	10.06 ± 0.58 aA	6.51 ± 0.16 bBC	6.54 ± 0.17 aBC	5.94 ± 0.15 aBCD
	Nymph stage									
F_0_	-	28.81 ± 0.27 aA	23.81 ± 0.27 aB	15.41 ± 0.49 aE	13.51 ± 0.47 aFG	12.025 ± 0.22 G	25.12 ± 0.31 aB	21.43 ± 0.55 aC	17.90 ± 0.26 bD	14.30 ± 0.09 aEF
F_1_	-	30.27 ± 0.54 aA	21.86 ± 0.39 bC	11.87 ± 0.52 bF	13.67 ± 0.28 aEF	-	25.42 ± 0.58 aB	19.84 ± 0.53 aD	17.69 ± 0.49 bD	14.24 ± 0.17 aE
F_2_	-	26.68 ± 0.40 bA	23.47 ± 0.34 aB	16.12 ± 0.51 aD	13.12 ± 1.32 aE	-	24.36 ± 0.36 aAB	17.34 ± 0.43 bCD	19.64 ± 0.33 aC	14.54 ± 0.55 aDE

^1^ Number of replication for each treatment (constant and fluctuating temperature) in each generation: F1 = 120, F2 = 320 and F3 = 100.

**Table 2 insects-14-00448-t002:** Degree day (dd) requirements from linear regressions of development rate (1/days within stage) vs. temperature. For dd models, the base temperature is indicated by the x-intercept, and dd requirements are indicated by 1/slope. Statistical values for regression, including r² and F tests for goodness of fit, are indicated. Note that the base temperature is a calculation requirement and NOT the biological minimum developmental temperature (see Higley and Haskell 2001 for details).

	Life Stage				
	2nd Instar (*n* = 120)	3rd Instar (*n* = 120)	4th Instar (*n* = 120)	5th Instar (*n* = 120)	2–5th Instar (*n* = 120)
Base temperature	14.1	15	15.2	13.5	15.5
Degree day requirement	51.2	43.7	47.8	84.9	197.4
r² of regression	0.7471	0.8071	0.8523	0.874	0.7521
F	171.3	242.7	334.7	402.3	175.9
df	1, 58	1, 58	1, 58	1, 58	1, 58
*p* value	<0.0001	<0.0001	<0.0001	<0.0001	<0.0001

**Table 3 insects-14-00448-t003:** Survival (%) of the nymphal stages of *Euschistus heros* submitted to different temperatures, constant and alternating, for three generations under laboratory conditions. Means followed by the same letter, lowercase in column (between generations) and uppercase in the line, do not differ from each other by Tukey test (α = 0.05).

Generation ^1^	Temperatures									
	Constant						Fluctuating			
	19 °C	22 °C	25 °C	28 °C	31 °C	34 °C	(25–21 °C)	(28–24 °C)	(31–27 °C)	(34–30 °C)
	2nd instar (A)									
F_0_	55.83 ± 4.23 C	77.50 ± 3.63 bAB	78.33 ± 5.17 bAB	79.16 ± 2.63 aAB	83.33 ± 3.48 aAB	90.00 ± 2.80 aA	80.83 ± 4.48 aAB	72.50 ± 5.36 abB	80.00 ± 3.06 abAB	89.99 ± 2.50 aA
F_1_	-	72.83 ± 4.86 bAB	87.05 ± 2.34 abA	48.47 ± 1.08 bC	70.83 ± 5.59 abAB	-	80.55 ± 3.51 aAB	64.66 ± 8.73 bBC	65.88 ± 9.06 bBC	59.44 ± 8.5 bBC
F_2_	-	99.16 ± 0.83 aA	99.37 ± 0.62 aA	81.04 ± 6.45 aB	62.50 ± 3.22 bC	-	86.66 ± 3.33 aAB	92.70 ± 3.86 aAB	95.41 ± 3.25 aAB	82.49 ± 0.83 aB
	3rd instar (B)									
F_0_	88.58 ± 3.75 A	88.83 ± 4.40 bA	95.08 ± 1.49 aA	91.91 ± 3.29 aA	90.49 ± 3.71 aA	89.33 ± 3.01 aA	86.08 ± 2.87 bA	89.50 ± 6.16 abA	91.25 ± 3.25 aA	86.99 ± 2.75 abA
F_1_	-	96.46 ± 1.96 abA	94.13 ± 2.25 aAB	96.94 ± 2.16 aA	81.41 ± 1.66 aABC	-	98.16 ± 1.63 aA	76.61 ± 5.92 bBC	72.98 ± 6.11 bC	65.83 ± 10.08 bC
F_2_	-	99.54 ± 0.45 aA	96.66 ± 3.33 aA	98.70 ± 1.29 aA	92.50 ± 5.00 aA	-	100.00 ± 0.00 aA	99.37 ± 0.62 aA	99.79 ± 0.20 aA	94.88 ± 3.87 aA
	4th instar (C)									
F_0_	56.24 ± 8.50 C	94.91 ± 2.28 bA	87.83 ± 2.47 bAB	83.66 ± 5.73 aAB	83.75 ± 3.06 bAB	74.25 ± 4.88 bcBC	88.58 ± 2.47 bAB	86.41 ± 2.57 aAB	85.58 ± 2.84 bAB	94.00 ± 3.10 aA
F_1_	-	100.00 ± 0.00 aA	92.52 ± 1.93 bA	97.77 ± 2.22 aA	89.00 ± 4.87 abA	-	95.22 ± 2.44 abA	95.66 ± 2.28 aA	88.72 ± 4.16 bA	88.58 ± 4.80 aA
F_2_	-	99.58 ± 0.41 abA	100.00 ± 0.00 aA	95.33 ± 3.26 aA	100.41 ± 0.41 aA	-	100.00 ± 0.00 aA	95.54 ± 3.91 aA	99.75 ± 0.25 aA	96.00 ± 4.00 aA
	5th instar (D)									
F_0_	9.00 ± 6.23 C	90.33 ± 3.09 bA	82.25 ± 6.00 aA	67.83 ± 7.39 bAB	79.75 ± 6.37 bA	47.50 ± 6.96 bB	90.08 ± 3.00 aA	85.50 ± 3.69 abA	79.08 ± 2.88 bA	73.66 ± 8.00 bA
F_1_	-	99.00 ± 1.00 aA	97.66 ± 1.45 aA	100.00 ± 0.00 aA	59.40 ± 5.47 bB	-	98.46 ± 1.29 aA	98.58 ± 1.31 aA	90.92 ± 3.86 aA	91.26 ± 3.91 abA
F_2_	-	98.70 ± 1.29 aAB	95.37 ± 3.89 aAB	94.06 ± 3.98 aAB	105.41 ± 7.08 aA	-	58.00 ± 6.63 bC	82.29 ± 5.60 bB	99.37 ± 0.62 aAB	100.00 ± 0.00 aAB
	Nymphal stage (E)									
F_0_	4.16 ± 3.39 C	59.16 ± 5.49 bA	54.16 ± 5.43 bA	41.66 ± 4.75 bAB	48.33 ± 5.03 aA	26.66 ± 3.89 bB	55.83 ± 1.66 bA	49.16 ± 2.76 bA	49.99 ± 4.56 bA	55.00 ± 6.64 abA
F_1_	-	69.61 ± 5.01 bA	75.00 ± 2.77 aA	45.83 ± 2.06 bB	30.00 ± 3.76 bB	-	73.83 ± 5.81 aA	48.38 ± 6.01 bB	41.83 ± 11.46 bB	36.66 ± 7.57 bB
F_2_	-	97.08 ± 2.91 aA	92.08 ± 6.46 aABC	70.83 ± 3.22 aCDE	60.00 ± 4.08 aDE	-	50.00 ± 5.27 bE	72.50 ± 6.40 aCD	94.37 ± 3.54 aAB	74.72 ± 3.73 aBCD

^1^ Number of replication for each treatment (constant and fluctuating temperature) in each generation: F1 = 120, F2 = 320 and F3 = 100.

**Table 4 insects-14-00448-t004:** Multiple decrement life table indicating impact of molting mortality on stage-specific and cumulative mortality: aqx = fraction of deaths from all causes in stage; alx = fraction of starting cohort alive at start of stage; adx = fraction of starting cohort deaths in stage.

Multiple Decrement Life Table				
Stage	Aqx	alx	adx	Fraction of Deaths by Molting
2nd instar	0.2122	1.0000	0.2122	0.2122
3rd instar	0.0899	0.7878	0.0708	0.0708
4th instar	0.0671	0.7170	0.0481	0.0481
5th instar	0.1179	0.6689	0.0789	0.0789
Adult	-	0.5900	-	-
Total			0.4100	0.4100

**Table 5 insects-14-00448-t005:** Pronotum size (mm) of adult *Euschistus heros* submitted to different temperatures, constant and alternating, for three generations under laboratory conditions. Means followed by the same letter, lowercase in column (between generations) and uppercase in the line, do not differ from each other by Tukey test (α = 0.05).

Generation ^1^	Temperatures									
	Constant						Fluctuating			
	19 °C	22 °C	25 °C	28 °C	31 °C	34 °C	(25–21 °C)	(28–24 °C)	(31–27 °C)	(34–30 °C)
	Male (A)									
F_0_	-	8.10 ± 0.15 bA	8.09 ± 0.06 aA	7.79 ± 0.11 aAB	7.86 ± 0.09 aAB	7.80 ± 0.13 AB	7.62 ± 0.08 aB	7.42 ± 0.09 bB	7.43 ± 0.05 bB	7.46 ± 0.07 abB
F_1_	-	8.46 ± 0.06 abA	8.27 ± 0.11 aA	8.10 ± 0.11 aAB	8.11 ± 0.10 aAB	-	7.38 ± 0.06 aC	7.70 ± 0.05 aBC	7.73 ± 0.08 aBC	7.41 ± 0.18 bC
F_2_	-	8.78 ± 0.02 aA	8.21 ± 0.04 aB	8.05 ± 0.02 aBC	8.01 ± 0.10 aBC	-	7.46 ± 0.06 aE	7.63 ± 0.06 abDE	7.86 ± 0.03 aCD	7.87 ± 0.06 aCD
	Female (B)									
F_0_	-	8.62 ± 0.12 bA	8.19 ± 0.04 cABC	8.34 ± 0.14 aAB	8.56 ± 0.08 aA	8.18 ± 0.10 ABC	7.83 ± 0.08 aCD	7.93 ± 0.09 aBCD	7.87 ± 0.08 abCD	7.71 ± 0.05 aD
F_1_	-	8.88 ± 0.07 abA	8.83 ± 0.11 aAB	8.65 ± 0.10 aAB	8.25 ± 0.15 aBC	-	7.53 ± 0.17 aD	8.03 ± 0.05 aCD	7.77 ± 0.12 bCD	7.67 ± 0.21 aCD
F_2_	-	9.10 ± 0.02 aA	8.49 ± 0.05 bBC	8.66 ± 0.04 aAB	8.60 ± 0.17 aBC	-	7.65 ± 0.20 aE	7.81 ± 0.05 aDE	8.21 ± 0.07 aBCD	8.14 ± 0.06 aCD

^1^ Number of replication for each treatment (constant and fluctuating temperature) in each generation: F1 = 120, F2 = 320 and F3 = 100.

**Table 6 insects-14-00448-t006:** Sex ratio of *Euschistus heros* submitted to different temperatures, constant and fluctuating, for three generations under laboratory conditions. Means followed by the same letter, lowercase in column and uppercase in the line, do not differ from each other by Tukey test (α = 0.05).

Generations ^1^	Temperatures								
	Constant					Fluctuating			
	22 °C	25 °C	28 °C	31 °C	34 °C	(25–21 °C)	(28–24 °C)	(31–27 °C)	(34–30 °C)
F_0_	0.47 ± 0.05 aA	0.53 ± 0.06 aA	0.56 ± 0.06 aA	0.43 ± 0.05 aA	0.63 ± 0.14 A	0.54 ± 0.03 Aa	0.54 ± 0.04 aA	0.52 ± 0.04 aA	0.48 ± 0.05 aA
F_1_	0.40 ± 0.04 aA	0.55 ± 0.02 aA	0.41 ± 0.11 aA	0.57 ± 0.07 aA	-	0.47 ± 0.04 aA	0.53 ± 0.05 aA	0.42 ± 0.05 aA	0.66 ± 0.10 aA
F_2_	0.55 ± 0.06 aA	0.49 ± 0.06 aA	0.49 ± 0.05 aA	0.48 ± 0.06 aA	-	0.61 ± 0.09 aA	0.48 ± 0.04 aA	0.50 ± 0.03 aA	0.44 ± 0.03 aA

^1^ Number of replication for each treatment (constant and fluctuating temperature) in each generation: F1 = 120, F2 = 320 and F3 = 100.

**Table 7 insects-14-00448-t007:** Pre-oviposition (d), number of eggs and viability (%) *Euschistus heros* submitted to different temperatures, constant and alternating, for three generations under laboratory conditions. Means followed by the same letter, lowercase column and uppercase in the line do not differ from each other by the Tukey test (*p* > 0.05).

Generation ^1^	Temperature									
	Constant						Fluctuating			
	19 °C	22 °C	25 °C	28 °C	31 °C	34 °C	(25–21 °C)	(28–24 °C)	(31–27 °C)	(34–30 °C)
	Pre-oviposition period (d)									
F_0_	-	11.03 ± 1.20 cA	7.67 ± 0.70 cABC	2.31 ± 0.41 cE	-	-	-	6.12 ± 1.00 cBCD	4.31 ± 0.52 cCDE	2.75 ± 0.33 cDE
F_1_	-	34.37 ± 0.48 bA	26.54 ± 1.00 bABC	16.80 ± 0.75 bD	-	-	-	29.23 ± 2.29 bAB	23.40 ± 3.93 bBCD	19.40 ± 1.12 bCD
F_2_	-	56.88 ± 0.91 aA	45.51 ± 0.38 aC	29.48 ± 0.45 aEF	-	-	-	47.26 ± 1.10 aBC	39.20 ± 1.11 aD	32.10 ± 0.57 aE
Generation	Number of eggs per female									
F_0_	-	291.15 ± 65.19 bAB	347.00 ± 57.97 bA	366.05 ± 107.50 aA	-	-	-	200.21 ± 38.56 aAB	203.53 ± 38.73 bAB	174.81 ± 40.96 aB
F_1_	-	552.48 ± 47.76 aA	534.31 ± 47.55 aA	356.80 ± 72.90 aABC	-	-	-	179.98 ± 64.89 aC	263.20 ± 84.26 abABC	197.20 ± 25.65 aC
F_2_	-	430.40 ± 15.26 abAB	560.52 ± 28.36 aA	371.27 ± 36.01 aAB	-	-	-	286.56 ± 48.78 aBC	426.14 ± 39.45 aAB	257.30 ± 49.72 aBC
Generation	Viability (%)									
F_0_	-	68.45 ± 7.53 aA	59.45 ± 3.25 bAB	65.21 ± 12.51 aA	-	-	-	29.51 ± 8.64 aBC	26.71 ± 8.63 aBC	15.56 ± 6.78 aC
F_1_	-	86.44 ± 2.83 aA	77.04 ± 3.25 aA	61.46 ± 6.72 aAB	-	-	-	8.77 ± 4.86 aC	16.08 ± 13.60 aBC	8.57 ± 8.57 aC
F_2_	-	72.17 ± 3.59 aA	67.42 ± 3.73 abAB	58.68 ± 5.64 aAB	-	-	-	22.44 ± 5.70 aC	33.23 ± 1.65 aBC	10.05 ± 6.22 aC

^1^ Number of replication for each treatment (constant and fluctuating temperature) in each generation: F1 = 120, F2 = 320 and F3 = 100.

## Data Availability

The data presented in this study are available on request from the corresponding author.

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
