# Peer review of "Trade-Offs between Temperature and Fitness in Euschistus heros (Fabricius) (Hemiptera: Pentatomidae): Implications for Mass Rearing and Field Management"

_insects, 2023, doi:10.3390/insects14050448_

Round 1

Reviewer 1 Report

This study provides some insights aimed at assessing the effects of constant and fluctuating temperatures on life characteristics of brown stink bug under lab conditions. In its current condition, however, I believe this manuscript is not yet acceptable for publication in the journal of Insects. I have provided summary based on my reading of the manuscript:

1) The introduction and discussion provide no insight on how this MS relates to the various other ones cited in the text or concerns that have been raised by other researchers. This article should provide details on all these fronts to provide the proper context for the work. Authors do not present any hypotheses or expectations that could be connected to previous studies; adding these details will improve the paper (see my comments below). The authors should clearly explain why the research was done, why it was important, and how it fits with other studies.

2) My primary concern is that the authors are extrapolating the applicability of their results beyond what the design supports. These are only data from a set of six highly artificial constant laboratory conditions and four fluctuating temperatures, so the inference power of the paper is very limited, but authors do not acknowledge this detail at all and need to be more forthcoming. The effect of fluctuating temperature profiles <16C and >34C on brown stink bugs was not investigated in this study. This is a critical limitation of the study, and the authors must concede and discuss this. The interaction of cyclic temperatures with nonlinear characteristics of brown stink bug development curves can introduce significant deviations from the results obtained here, and especially at the lower and higher temperatures of development functions, such as <16C and >34C. Studies across a broader set of constant and especially fluctuating temperature regimes are therefore encouraged so that more realistic effect of temperature on biological parameters of brown stink bugs could be elucidated, as this is the closest to temperature fluctuations that occur in the field. So, I am suggesting to the authors to tone-down the language a little and admit that there are still substantive uncertainties to be considered.

Some of the authors’ statements would be much stronger if they tie their work to the body of literature that has built up on the bio ecology of hymenopteran biocontrol agents (BCAs), e.g., Journal of Economic Entomology 112:1560-1574 and Journal of Economic Entomology 112:1062-1072. These studies provide strong evidence that daily temperature fluctuations significantly affected development times and longevity of BCAs, resulting in marked deviations and potentially erroneous predictions when compared to their constant temperature regimen counterparts. In those studies, each fluctuating temperature profile was modeled after field recorded temperatures that had the desired average target temperature. These are the first studies ever to undergo such analysis. This article should provide details on all these fronts to provide the proper context for the work. This is not to diminish the data gathered in this study, as they are of value. But it is important for the authors not to overgeneralize, and to warn the reader, including regulatory agencies, against doing so as well. Adding these details will improve the discussion.

3) Also, the discussion lacks real concluding remarks in my opinion, and if I was a practitioner or consultant, I’d want to see these recommendations for my area or city. The conclusions should also concisely summarize major findings and suggest, briefly, new avenues for research.

Overall, I was excited to see the results of the paper after reading the abstract, but I found it hard to extract key messages useful to policymakers and professionals, probably in large part due to the lack of connection with other published work and need for improved structure of the current manuscript.

Good luck!

Author Response

Reviewer #1

  • The introduction and discussion provide no insight on how this MS relates to the various other ones cited in the text or concerns that have been raised by other researchers. This article should provide details on all these fronts to provide the proper context for the work. Authors do not present any hypotheses or expectations that could be connected to previous studies; adding these details will improve the paper (see my comments below). The authors should clearly explain why the research was done, why it was important, and how it fits with other studies

 My primary concern is that the authors are extrapolating the applicability of their results beyond what the design supports. These are only data from a set of six highly artificial constant laboratory conditions and four fluctuating temperatures, so the inference power of the paper is very limited, but authors do not acknowledge this detail at all and need to be more forthcoming. The effect of fluctuating temperature profiles <16C and >34C on brown stink bugs was not investigated in this study. This is a critical limitation of the study, and the authors must concede and discuss this. The interaction of cyclic temperatures with nonlinear characteristics of brown stink bug development curves can introduce significant deviations from the results obtained here, and especially at the lower and higher temperatures of development functions, such as <16C and >34C. Studies across a broader set of constant and especially fluctuating temperature regimes are therefore encouraged so that more realistic effect of temperature on biological parameters of brown stink bugs could be elucidated, as this is the closest to temperature fluctuations that occur in the field. So, I am suggesting to the authors to tone-down the language a little and admit that there are still substantive uncertainties to be considered.

Some of the authors’ statements would be much stronger if they tie their work to the body of literature that has built up on the bio ecology of hymenopteran biocontrol agents (BCAs), e.g., Journal of Economic Entomology 112:1560-1574 and Journal of Economic Entomology 112:1062-1072. These studies provide strong evidence that daily temperature fluctuations significantly affected development times and longevity of BCAs, resulting in marked deviations and potentially erroneous predictions when compared to their constant temperature regimen counterparts. In those studies, each fluctuating temperature profile was modeled after field recorded temperatures that had the desired average target temperature. These are the first studies ever to undergo such analysis. This article should provide details on all these fronts to provide the proper context for the work. This is not to diminish the data gathered in this study, as they are of value. But it is important for the authors not to overgeneralize, and to warn the reader, including regulatory agencies, against doing so as well. Adding these details will improve the discussion.

We thank the reviewer for valuable feedback. We acknowledge the concerns raised and addressed them in our revisions to provide a more comprehensive and balanced perspective on our research.

We added the sentence in Introduction:

“ Prior research has examined the influence of fluctuating temperatures on a broad range of diverse insects, including biological control agents[21,22], aquatic insects [23] and the brown marmorated stink bug, Halyomorpha halys (StaÌŠl, 1855) (Hemiptera: Pentatomidae) [24]. However, for the brown stink bug, no scientific literature has investigated its development and reproduction under fluctuating temperature conditions.”

And a last statement in Discussion:

“Further tests to confirm these results are warranted. All of our experiments were conducted under well-controlled laboratory conditions, which may not suitably represent the complex and dynamic environmental factors that E. heros experiences in the field. In the field, it is likely that adults seek microclimate refuges and thus, laboratory results under constant conditions may not represent the behavior and development of the stink bug under natural conditions. Further studies are necessary to examine other abiotic factors such as humidity, precipitation, and varying light regimes, which might also influence the development and survival of the brown stink bug. Furthermore, future research should also address the effect of fluctuating temperatures on its natural enemy, the egg parasitoid T. podisi, to provide a better optimize its potential for biological control programs.”

  • Also, the discussion lacks real concluding remarks in my opinion, and if I was a practitioner or consultant, I’d want to see these recommendations for my area or city. The conclusions should also concisely summarize major findings and suggest, briefly, new avenues for research.

A conclusion was added:

“The present study found that there is no development at constant temperatures of 19°C and 34°C, and no reproduction at 31°C. An inverse relationship was observed between temperature and nymphal development duration at remaining temperatures. The study also found that temperature had a significant impact on the survival and pre-oviposition period of E. heros. Furthermore, it provided degree-day requirements for each instar, and identified molting as a critical factor in population mortality (more than 40%), , shedding light on an important factor of E. heros natural mortality, the ecdysis process. Despite limitations of the study, these findings have important implications for E. heros laboratory mass rearing programs, guiding the adoption of better temperature regimes to produce eggs for mass rearing egg parasitoids and for its management in the field.”

Reviewer 2 Report

This is a potentially good paper targeting development speed, degree days requirements, mortality, egg production, body size and other baseline-but-crucial physiology parameters related to rearing temperatures, and their fluctuations, on the pentatomid bug Euschistus heros, a pest of soya crops in South America. 

Publishing such results is of utmost importance in pest control science, especially - as the authors rightly explain - because mass rearing the bug is necessary for commercial production of its hymenopteran parasitoid,  Telenomus podisi, used in crop protection. Last but not least, producing data for the manuscript took a lot of diligent laboratory work, which is admirable.     
So much being said, the Msc needs considerably more desktop work to be publishable, both regarding the style, language, and the presentation of results. My specific comments will be marked as "MAJ" and "MIN", but both categories are important. Note, however, that there are much more "MIN" issues than those which I will directly comment; just use my MIN comments as a guide, and find+correct the others yourself. 

Abstract, MAJ

When mentioning the bugs name for the first time, full scientific name, i.e. including the year of description, must be stated. The same apply, below at line 68 of Introduction, for the parasitioid, when mentioned for the first time.

Introduction, MAJ

For general reader not vested in your system, more of basic information on the targeted species will make the text more readable. Specifically, you mention that the bug was "umcommon" in Neotropical region; so where did it come from, is it perhaps an alien/invasive species? How is its life cycle in natural conditions, and by which mechanisms does it impair the crops? 

line 58, MAJ. "...when there are two adult stink bugs and 3rd to 5th stage nymphs per meter..." Is it really meter, or meter-squared? Also, is there really "and", or "or", as these may mean different things, in terms of the bugs quantity. 

68, MAJ - regarding the parasitoid Telenomus podisi: family, full scientific name, and perhaps geographic origin will make your story more interesting. 

Material and methods

MAJ, lines 83-90

You described in detail the rearing conditions, which is OK, but how did you get the bugs for rearing? Were there wild-caught, did they originate from a captive population, which stage did you start with, how many females/mothers were used to obtain the first generation eggs ??? Please, elaborate. 

MAJ, lines 106, and below

I do not really unerstand the term "pseudo-replicates", as this can signify many different things, from having all the rearing containers with a temperature/fluctuation treatment in a single box ( = completely understandable, as perfect independent replications are prohibitively costly in such a case), complex family/origin effects on origin of the material - and many others. I praise your honesty in selecting this term, but an explanation is necessary. Perhaps a sentence, such as, "Due to high financial and spatial cost of perfect replication, we .... and hence, our replicates represent pseudoreplications, which... (are used in most of studies similar to ours)" 

MIN, around line 108

As the numbers of living bugs decreased from instar to instar, 

MIN, 147 - 50 
The error distributions used in statistical tests. Although I understand why you used these somehow exotic distributions, other readers may not. Please, explain (in terms of deviance, degrees of freedom, etc.). Perhaps, for easier replicability, you may directly present the calls you used in R to produce the two categories of models, if not more detailed regreession results in a Supplementary material. 

RESULTS, 
MAJ, general

It is a good practice in scientific writing not to present the same information in Tables and Figures, which you do, repeatedly, e.g., in Table 1 and Figure 1. I understand why you opted to do so - the table allows you to present SEs of the estimates, the values for treatments which died off, etc. But still, could not you present the errors in the figures, somehow? Give it some effort, it will pay by making the paper more readable, as at this stage, it is rather a student's lab protocol than a paper for others to read. 

Tables, MIN, general

perhaps, the numbers of replications per instar might be stated directly in the tables, sparing the readers from back-checking with methos.

Figure 3, MIN

What are the box-plots showing, are these SEs, SDs, or which measures of variation in the data? Please, indicate. 

MIN, lines 179-180
an example of expression, which can be made clearer by just use of simpler words; there are many such cases in the text, please, go through it and do some editing: "The total degree day requirement for nymphal development was 197.4 degree days, with the 5th instar requiring more highest quantity of degree days (84.9 degree days) and the 3rd instar requiring the lowest (43.7 degree days; table 2)" => "The total degree day requirement for nymphal development was 197.4 degree days. The 5th instar required the degree days number (84.9), the 3rd instar required the lowest number (43.7; table 2)."

MIN, throughout
Consider establishing and using the commonly used shortcut "dd" for degee-days. 

MAJ, lines 236 and 247
What happened with tables 7+8? I cannot see them there. 

Generally, regarding the Results, the combined number of tables and figures is just huge. I believe that you may reduce them by avoiding the repetitive information in tables and figures. 

Discussion

I must admit that I read it only superficially, because - given the missing life history and area of origin information in Introduction - I cannot really decipher what do your findings imply for general biology of the studied species. For instance, you found that 19 C degrees was too low, and 34 C degrees too high temperature for rearing the bugs. Does it mean anything, in terms of either origin/biogeography of the species, or its control? The same may hold for fluctuating vs. constant temperatures; knowing more about the insect in its native habitats might provide material for interesting confrontations. 

line 257, MIN

Provide full scientific name, and perhaps family, for Halyomorpha halys. 

around line 264, MIN

The increased preoviposition period in F1, F2: Can you give some advice for future workers, how to minimize this effect in rearing the bug?  

It seems to be OK in terms of grammar, but more editing is necessary to make the text more concise, and hence readable. 

Author Response

  • Abstract, MAJ

When mentioning the bugs name for the first time, full scientific name, i.e. including the year of description, must be stated. The same apply, below at line 68 of Introduction, for the parasitioid, when mentioned for the first time.

Authors: Done!

  • Introduction, MAJ

For general reader not vested in your system, more of basic information on the targeted species will make the text more readable. Specifically, you mention that the bug was "umcommon" in Neotropical region; so where did it come from, is it perhaps an alien/invasive species? How is its life cycle in natural conditions, and by which mechanisms does it impair the crops?

The sentences were inserted in order to address this suggestion:

“The brown stink bug utilizes its stylets to feed on various structures of the soybean plant, but its feeding on the pod can have significant consequences for soybean production, including reduced yield, altered grain quality parameters, and changes in oil and protein content of the seeds during pod development and seed filling [7,8]. In the field, E. heros undergoes five nymphal instars, but its damage to soybeans is only significant after the third instar [4,9].”

  • line 58, MAJ. "...when there are two adult stink bugs and 3rd to 5th stage nymphs per meter..." Is it really meter, or meter-squared? Also, is there really "and", or "or", as these may mean different things, in terms of the bugs quantity.

The recommended sampling method for soybean in Brazil is the vertical beat sheet, which is 1.0 meter long, and the ET for stink bugs on soybean is based on this method. Therefore, to avoid any misunderstandings, we changed the text:

“Despite the well-established ET for brown stink bugs on soybean, which recommends treatment when there are two stink bugs per meter (3rd to 5th instar nymphs and/or adults) [4,7], soybean growers still express concerns about stink bug population growth in the field, particularly when numerous eggs and small nymphs up to the 2nd instar are detected during sampling.

  • 68, MAJ - regarding the parasitoid Telenomus podisi: family, full scientific name, and perhaps geographic origin will make your story more interesting.

Done.

  • Material and methods - MAJ, lines 83-90

You described in detail the rearing conditions, which is OK, but how did you get the bugs for rearing? Were there wild-caught, did they originate from a captive population, which stage did you start with, how many females/mothers were used to obtain the first generation eggs ??? Please, elaborate.

The following sentence was added: “The laboratory rearing of E. heros originated from insects collected in a soybean field at the Experimental Farm Lageado of the Faculty of Agronomic Sciences of UNESP-Botucatu, São Paulo State, Brazil (geographic coordinates 22°48'19.5"S 48°25'38.4"W). The founding unit of the rearing was comprised of 156 insects, including 70 males and 86 females, and these insects were maintained for three generations until the beginning of the experiment.”

  • MAJ, lines 106, and below

I do not really understand the term "pseudo-replicates", as this can signify many different things, from having all the rearing containers with a temperature/fluctuation treatment in a single box ( = completely understandable, as perfect independent replications are prohibitively costly in such a case), complex family/origin effects on origin of the material - and many others. I praise your honesty in selecting this term, but an explanation is necessary. Perhaps a sentence, such as, "Due to high financial and spatial cost of perfect replication, we .... and hence, our replicates represent pseudoreplications, which... (are used in most of studies similar to ours)"

The followed sentence was included in order to explain the term pseudo-replication:

“The experimental design comparing temperatures was pseudo-replicated, as the experimental unit was the chamber, not the individual petri dishes. As a result, a conservative approach was taken when making treatment comparisons.”

  • MIN, around line 108

As the numbers of living bugs decreased from instar to instar,

We clarified the reason for a change in treatments by adding “Because E. heros was unable to survive and reproduce at all tested temperatures, the number of pseudo-replications (n) was based on results from the previous generation.”

  • MIN, 147 - 50

The error distributions used in statistical tests. Although I understand why you used these somehow exotic distributions, other readers may not. Please, explain (in terms of deviance, degrees of freedom, etc.). Perhaps, for easier replicability, you may directly present the calls you used in R to produce the two categories of models, if not more detailed regression results in a Supplementary material.

Regarding the probability distributions, we provided a brief explanation about the type of data for each residual probability distribution. In this way, the current format of the manuscript is still valuable, as it allows any interested reader to contact the authors for further explanations, as has occurred with other articles. After explanation, we do not believe presentation of the statistics as a Supplementary material is justified.

“The variable's survival, sex ratio and viability of the eggs were analyzed using a generalized linear model (GLM) with residual Bernoulli distribution, since data vary between 0 and 1 for sex ratio and from 0 to 100 for survival and viability of the eggs. As for the variable total number of eggs, it was analyzed using a GLM with negative binomial residual distribution, since these are counting data. Analyzes were performed using the ExpDes.pt package in the R computing environment [22].”

  • RESULTS, MAJ, general

It is a good practice in scientific writing not to present the same information in Tables and Figures, which you do, repeatedly, e.g., in Table 1 and Figure 1. I understand why you opted to do so - the table allows you to present SEs of the estimates, the values for treatments which died off, etc. But still, could not you present the errors in the figures, somehow? Give it some effort, it will pay by making the paper more readable, as at this stage, it is rather a student's lab protocol than a paper for others to read.

We agree.  In response, Tables 5, 7 and 8 and Figures 1, 2 and 5 were removed.

  • Tables, MIN, general

perhaps, the numbers of replications per instar might be stated directly in the tables, sparing the readers from back-checking with methos.

Done.

  • Figure 3, MIN

What are the box-plots showing, are these SEs, SDs, or which measures of variation in the data? Please, indicate.

An explanation of what a box-plot means is now provided in the footnote.

  • MIN, lines 179-180

an example of expression, which can be made clearer by just use of simpler words; there are many such cases in the text, please, go through it and do some editing: "The total degree day requirement for nymphal development was 197.4 degree days, with the 5th instar requiring more highest quantity of degree days (84.9 degree days) and the 3rd instar requiring the lowest (43.7 degree days; table 2)" => "The total degree day requirement for nymphal development was 197.4 degree days. The 5th instar required the degree days number (84.9), the 3rd instar required the lowest number (43.7; table 2)."

Thank you for the suggestion. The changes were made.

  • MIN, throughout

Consider establishing and using the commonly used shortcut "dd" for degee-days.

The abbreviation “dd” was adopted throughout.

  • MAJ, lines 236 and 247

What happened with tables 7+8? I cannot see them there.

Considering the suggestion number 15, we removed some tables and figures, including the recently added Table 7. Table 8 was inserted and renamed as "Table 7."

  • Generally, regarding the Results, the combined number of tables and figures is just huge. I believe that you may reduce them by avoiding the repetitive information in tables and figures.

Tables 5, 7 and 8 and Figures 1, 2 and 5 were removed in order to avoid repetitive information.

  • Discussion

I must admit that I read it only superficially, because - given the missing life history and area of origin information in Introduction - I cannot really decipher what do your findings imply for general biology of the studied species. For instance, you found that 19 C degrees was too low, and 34 C degrees too high temperature for rearing the bugs. Does it mean anything, in terms of either origin/biogeography of the species, or its control? The same may hold for fluctuating vs. constant temperatures; knowing more about the insect in its native habitats might provide material for interesting confrontations.

Thank you for the insightful comments. We addressed these comments throughout the manuscript.

  1. heros is native to Neotropical regions, specifically in South America and Panama. This manuscript presents important information about stink bug natural death due to molting, with a mortality rate of more than 40%. Half of this occurs at the 2nd instar (a stage not considered for ET). In practical terms, this implies that when a grower finds a mass of eggs or a considerable number of 1st and 2nd instar stink bugs during sampling in the field, they must consider that not all of them will reach the 3rd instar and cause significant damage to soybean yield. Additionally, the remaining nymphs will be exposed to other factors that can cause a decrease in their population, such as natural enemies and unfavorable climatic conditions.

  • line 257, MIN

Provide full scientific name, and perhaps family, for Halyomorpha halys.

Done!

  • around line 264, MIN

The increased preoviposition period in F1, F2: Can you give some advice for future workers, how to minimize this effect in rearing the bug? 

We provided advice on rearing based on our results. We recommend to avoid the temperature of 28°C for mass laboratory mass rearing. At temperatures of 22 and 25°C although it also increases the preoviposition period, it also increased the number of eggs per female, at F1 and F2 comparing to the F0 generation.

Round 2

Reviewer 1 Report

Authors have done a nice job addressing all of my original comments and those of other reviewers. I have no further suggestions to improve the paper. Thank you!

Reviewer 2 Report

The authors have done good job while reviewing the paper, I have no further issues with it and recommend its publication.